# Validation-Gated Hebbian Learning for Adaptive Agent Memory

**Pragya Singh**[*]
University of Pennsylvania
pragya7@seas.upenn.edu

**Stanley Yu**[*]
University of Pennsylvania
stany@seas.upenn.edu

## Abstract

LLM-based agents struggle with catastrophic forgetting, context limitations, and reasoning drift. While knowledge graphs (KGs) offer structured memory, current implementations remain static and do not adapt based on reasoning effectiveness. We introduce Kairos, a multi-agent reasoning system implementing Hebbian plasticity mechanisms for adaptive knowledge graphs. Kairos formalizes three neuroplasticity-inspired operations: edge strengthening (LTP analog), temporal decay (LTD analog), and emergent connection formation. A key innovation is validation-gated learning, where graph consolidation only occurs when reasoning passes multi-dimensional quality assessment (logical, grounding, novelty, alignment), preventing hallucination reinforcement. Our controlled proof-of-concept demonstrates that validation-gated Hebbian learning is both mechanically sound and practically beneficial, with adaptive graphs outperforming static baselines. We additionally identify a design principle showing that novelty and correctness are orthogonal dimensions that degrade when averaged in validation systems. These results establish the feasibility of adaptive agent memory where knowledge structures improve through validated iteration.

## 1 Introduction

Large language model (LLM)-based agents face persistent challenges with long-term memory and reasoning stability, hindered by catastrophic forgetting [Li et al., 2024, Luo et al., 2023], context window limitations [Packer et al., 2023], and reasoning drift [Chen et al., 2023]. Expanded context windows introduce quadratic costs and the "lost in the middle" effect where information is ignored [Packer et al., 2023].

Knowledge graphs (KGs) provide structured representations for complex, multi-hop reasoning [Pan et al., 2024], offering a more robust foundation than raw context accumulation. However, a critical gap remains: current systems treat graphs as static databases. While graphs may grow through data ingestion or agents may adapt navigation strategies, the underlying structure rarely learns from reasoning outcomes. Connections are not strengthened by successful use, and ineffective pathways do not weaken.

Biological memory offers inspiration. Synaptic connections strengthen through repeated co-activation through the Hebbian principle of "neurons that fire together wire together" [Hebb, 1949, Squire et al., 2015]. This suggests KGs could evolve based on reasoning utility, with structure optimized through validation-based learning rather than generative expansion alone [Buehler, 2025].

We present Kairos, a multi-agent reasoning system implementing Hebbian plasticity mechanisms for knowledge graphs. Successful reasoning paths are strengthened (long-term potentiation), unused

---

[*]Equal Contribution

edges weaken (long-term depression), and frequently co-activated concepts form emergent connections. Critically, this adaptation is *validation-gated*: only reasoning passing multi-dimensional quality assessment triggers consolidation, preventing hallucination reinforcement.

Our contributions are: 1) A validation-gated learning architecture where graph consolidation is conditioned on multi-dimensional quality assessment, preventing hallucination reinforcement while enabling adaptive memory. 2) Formalization of three neuroplasticity-inspired mechanisms (edge strengthening, temporal decay, emergent connections) for symbolic KG structures in multi-agent systems. 3) A multi-agent validation framework with specialized agents assessing complementary quality dimensions (logical consistency, factual grounding, novelty, alignment). 4) Demonstration through controlled evaluation that these design choices yield both mechanical correctness and practical utility, with adaptive graphs outperforming static baselines.

## 2 Related Work

### 2.1 Graph-Based Retrieval and Reasoning Systems

Graph-based retrieval augmentation has evolved significantly from early semantic networks. Microsoft's GraphRAG [Edge et al., 2024] introduced hierarchical summarization for query-focused retrieval, inspiring efficiency optimizations in LightRAG [Guo et al., 2024] and neurobiologically-inspired indexing in HippoRAG [Bae et al., 2024]. Reasoning-over-graphs approaches [Luo et al., 2024] enable multi-hop inference through traversal, while hybrid systems [Sarmah et al., 2024] combine semantic and structured search. Plan-on-Graph [Chen et al., 2024] introduced adaptive query planning. These advances focus on retrieval optimization and navigation rather than structural adaptation from reasoning feedback.

### 2.2 Agent Memory Architectures

Agent memory systems balance storage capacity with retrieval efficiency. MemGPT [Packer et al., 2023] pioneered hierarchical memory management, extended by A-Mem [Xu et al., 2025] with atomic linkable units. Generative Agents [Park et al., 2023] demonstrated importance-weighted retrieval combining recency and relevance. Classical cognitive architectures, such as ACT-R's activation-based consolidation [Anderson et al., 2004] and Soar's chunking mechanisms [Laird, 2012], implement usage-driven adaptation but remain disconnected from modern neural approaches. Multi-agent frameworks [Liang et al., 2023, Zhang et al., 2024] leverage KGs primarily for coordination rather than adaptive memory.

### 2.3 Dynamic Knowledge Graphs and Continual Learning

Knowledge graph dynamics typically respond to external data rather than internal reasoning. Temporal approaches like Know-Evolve [Trivedi et al., 2017] model event-driven updates through point processes, while LLM-DA [Wang et al., 2024] adapts temporal rules from language understanding. Emergent graph expansion [Buehler, 2025] generates new structures during reasoning. Continual learning methods [Daruna et al., 2021] accommodate new entities without catastrophic forgetting. These methods adapt content but not connection strength based on reasoning utility.

### 2.4 Neuroscience-Inspired Learning Mechanisms

Biological memory consolidation provides computational metaphors for adaptive systems. Hebbian plasticity, and specifically long-term potentiation strengthening co-activated synapses and depression weakening unused connections [Hebb, 1949, Squire et al., 2015], has inspired graph neural network architectures [Liu et al., 2024]. Complementary mechanisms include synaptic scaling for homeostatic stability [Turrigiano, 2008] and systems consolidation transferring episodic to semantic memory [Dudai, 2004]. Neural-symbolic AI research [Yang et al., 2025] identifies dynamic rule learning as an open challenge for integrating adaptive mechanisms with symbolic structures.

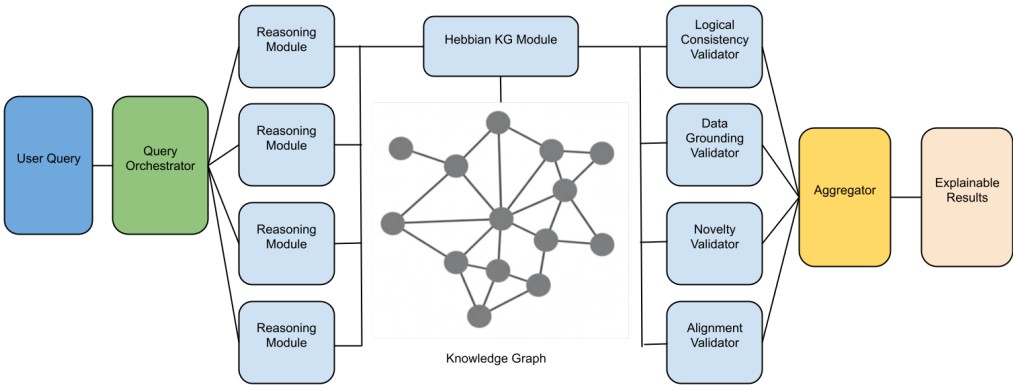

Figure 1: The Kairos system architecture. A user query is processed by an orchestrator, specialized reasoning modules, and a multi-agent validation layer. These components interact with a central knowledge graph, which is dynamically updated by a Hebbian KG module.

## 2.5 Positioning Kairos

Kairos contributes validation-gated structural adaptation: knowledge graphs evolving through reasoning effectiveness rather than data ingestion or navigation optimization. By implementing neuroplasticity-inspired mechanisms (edge strengthening, temporal decay, emergent connections) we enable episodic reasoning patterns to consolidate into semantic structure. Our proof-of-concept establishes technical feasibility and identifies critical design principles for multi-dimensional validation, particularly the orthogonality of novelty and quality assessment.

# 3 System Architecture

## 3.1 Core Components

**Query Orchestrator.** The orchestrator serves as the system's entry point, receiving user queries and routing them to appropriate reasoning modules. Module selection employs semantic similarity computed via sentence transformers (all-MiniLM-L6-v2), matching query embeddings against module descriptions. For complex queries requiring multiple perspectives, the orchestrator can invoke several modules sequentially or in parallel.

**Specialized Reasoning Modules.** Kairos employs a modular architecture with domain-specific agents. Throughout this, we will use the case study of crypto analysis, which has specialized agents for tasks like security auditing and financial analysis. This demonstrates flexibility across rule-based, data-driven, and LLM-augmented reasoning paradigms.

Each module queries the knowledge graph to retrieve relevant entities and relationships, constructs a reasoning path explaining its inference process, and produces a structured output containing: (1) a step-by-step reasoning path with data sources and logical inferences, (2) a conclusion with confidence score, (3) the specific KG triples used (source_triples), and (4) relevant metrics. This structured output enables downstream validation and Hebbian learning by making explicit which graph elements contributed to the reasoning.

**Dynamic Knowledge Graph.** The knowledge graph represents information as entity-relation triples with rich metadata. Each relation stores not only subject-predicate-object structure but also a confidence score (0.0-1.0), source provenance, temporal versioning, and Hebbian-specific metadata including activation count and cycles since last activation. This metadata enables the system to track usage patterns over usage. The graph supports both static triples extracted during document ingestion and emergent relations formed through co-activation patterns during reasoning.

**Multi-Agent Validation Layer.** Four specialized validation agents assess reasoning quality from complementary perspectives before any graph adaptation occurs. This diversity of validation helps

prevent premature consolidation of flawed reasoning patterns [Xu et al., 2025]. The validation layer outputs numerical scores (0-1) and textual feedback for each dimension, which gate the Hebbian learning process.

**Aggregator and Results.** The aggregator synthesizes outputs from multiple reasoning modules and validation agents, computing aggregate trust scores and presenting explainable results with provenance tracking to the user.

## 3.2 Validation-Gated Learning Feedback Loop

The critical architectural feature distinguishing Kairos from prior work is the validation gate between reasoning and learning. When a reasoning module produces output, validation agents assess quality across four dimensions (detailed in Section C). Only when *all* validators indicate acceptable quality (scores above threshold) does the system trigger Hebbian updates to strengthen the edges traversed during reasoning. This gate serves two purposes: (1) it prevents consolidation of hallucinated facts or logically incoherent reasoning paths, and (2) it aligns with neuroscientific findings that successful task completion, not mere neural activity, drives synaptic strengthening [Squire et al., 2015]. Failed reasoning attempts do not trigger strengthening; they simply fade through temporal decay, mirroring biological learning.

# 4 Hebbian Plasticity for Knowledge Graphs

## 4.1 Motivation: Beyond Static Knowledge Organization

Kairos formalizes neuroplasticity mechanisms for KG-based agent memory through three operations: edges traversed during *validated* reasoning strengthen (LTP analog), unused edges weaken over time (LTD analog), and frequently co-activated entities form emergent connections. Our evaluation confirms these mechanisms operate as specified and identifies critical design principles for multi-dimensional validation systems.

## 4.2 Mechanisms

### 4.2.1 Edge Strengthening: Long-Term Potentiation Analog

When a reasoning module's output passes validation, the KG edges listed in its `source_triples` field receive a strengthening signal. We implement asymptotic strengthening with diminishing returns:

$$\Delta_{\text{strength}} = \eta \times (\text{max\_strength} - \text{current\_strength}) \tag{1}$$

$$\text{new\_strength} = \min(\text{max\_strength}, \text{current\_strength} + \Delta_{\text{strength}}) \tag{2}$$

where $\eta = 0.1$ is the learning rate and max_strength $= 1.0$. This formulation mirrors biological LTP [Squire et al., 2015], allowing for noticeable adaptation within 10-20 episodes while preventing single-trial over-consolidation. For multi-hop paths, each edge receives proportional strengthening.

### 4.2.2 Temporal Decay: Long-Term Depression Analog

Edges not traversed during reasoning gradually weaken via temporal decay, analogous to synaptic depression [Squire et al., 2015]. We implement exponential decay:

$$\text{decay} = \gamma \times \left(1 - \exp\left(-\frac{\text{cycles\_inactive}}{\lambda}\right)\right) \tag{3}$$

$$\text{new\_strength} = \max(\text{min\_strength}, \text{current\_strength} - \text{decay}) \tag{4}$$

where $\gamma = 0.05$ is the decay rate, $\lambda = 5$ is the half-life in reasoning cycles, and min_strength $=$ 0.1 is a pruning threshold. This mechanism balances knowledge retention against removal of

irrelevant associations, adapting to actual usage patterns rather than wall-clock time while preventing catastrophic forgetting.

### 4.2.3 Emergent Connection Formation

Kairos also forms emergent relationships by tracking entity co-activations. When two entities appear together in reasoning contexts $N = 3$ or more times, a new `co_occurs_with` edge is created:

$$\text{initial\_strength} = \min(0.5, \text{count} \times 0.1) \tag{5}$$

This discovers implicit, cross-domain connections. The threshold $N = 3$ reduces noise, and the initial strength is capped at 0.5 to distinguish empirical edges from source-derived facts.

**Example:** If entities `Security-Audit` and `High-Risk` are co-activated three times across different queries in our blockchain analysis domain, Kairos creates an emergent edge:

```
Security-Audit -[co_occurs_with, strength=0.3]-> High-Risk
```

This new connection captures an empirical pattern that can accelerate future reasoning within the domain.

$$\text{trigger\_hebbian} \leftarrow \begin{cases} \text{True} & \text{if all validators pass: } v_i.\text{valid} = \text{True} \\ \text{False} & \text{otherwise} \end{cases} \tag{6}$$

where each validator $v_i$ produces a binary validity decision. Edge strengthening and entity co-activation tracking occur unconditionally during reasoning, but consolidation operations (emergent edge formation and temporal decay) only execute when all validators indicate successful reasoning. This implements consolidation inspired by reward-modulated plasticity in biology [Squire et al., 2015].

## 5 Results

We evaluate Kairos as a proof-of-concept system through three complementary experiments: (1) **Mechanical Validation**: Confirming Hebbian mechanisms operate as designed by tracking edge strength evolution over repeated reasoning cycles, (2) **Utility Validation**: Demonstrating that adaptive graphs outperform static baselines through direct comparison; and (3) **Architectural Analysis**: Examining the contribution of each system component through ablation study. Our evaluation uses a minimal knowledge graph representing a blockchain security audit scenario (ApolloContract smart contract with known vulnerabilities).

### 5.1 Experimental Setup

**Evaluation Dataset.** We constructed a 60-question evaluation dataset covering diverse query types: security audits, risk analysis, multi-hop reasoning, counterfactual scenarios, and meta-reasoning tasks. Questions vary in complexity from simple entity lookups (*"Has ApolloContract been audited?"*) to complex synthesis tasks (*"What is the holistic assessment of deploying smart contracts in the current environment?"*). This minimal setup serves as a controlled proof-of-concept for the Hebbian mechanisms.

**Metrics.** We measure: (1) *Trust score*: average of four validation dimensions (0-1 scale), serving as an aggregate quality metric; (2) *Individual validation scores*: logical coherence, factual grounding, novelty, and alignment (each 0-1); (3) *Hebbian metrics*: edges strengthened, entities activated, emergent connections formed; (4) *Edge strength*: confidence values of frequently-traversed graph edges (0-1).

### 5.2 Mechanical Validation: Hebbian Plasticity Over Time

To evaluate whether Hebbian mechanisms operate as designed, we run 5 reasoning cycles with the same set of 3 queries repeated in each cycle. We track edge strength evolution for frequently-traversed paths. Results are shown in Table 1.

Table 1: Hebbian plasticity evaluation across 5 reasoning cycles (queries per cycle: 3, 3, 1, 2, 3). Edge strength increases as frequently-used paths are reinforced.

| Cycle | Trust Score | Avg Edge Strength | Edges Strengthened |
|---|---|---|---|
| 1 | $0.708 \pm 0.052$ | $0.919 \pm 0.009$ | 3 |
| 2 | $0.683 \pm 0.063$ | $0.941 \pm 0.006$ | 3 |
| 3 | 0.700 | 0.957 | 1 |
| 4 | $0.775 \pm 0.035$ | $0.967 \pm 0.003$ | 2 |
| 5 | $0.750 \pm 0.075$ | $0.977 \pm 0.002$ | 3 |
| $\Delta$ (Cycle 5 vs 1) | +5.9% | +6.3% | — |

**Edge Strengthening Confirmation.** The average strength of frequently-used edges increases monotonically from $0.919$ (cycle 1) to $0.977$ (cycle 5), a $6.3\%$ gain. This confirms the Hebbian mechanism operates according to the asymptotic strengthening formula (Eq. 1): with learning rate $\eta = 0.1$, edges approach maximum strength (1.0) gradually through repeated activations. The mechanical behavior matches the designed specification.

**Temporal Decay Confirmation.** While emergent connections did not form due to minimal graph structure, temporal decay operated as designed. Edges not traversed in later cycles showed strength reduction consistent with the exponential decay formula (Eq. 3-4). With $\lambda = 5$ cycles, unused edges exhibited decay behavior confirming the mechanism functions according to specification.

**Emergent Connections.** No emergent connections formed during this evaluation. Given the minimal graph structure and the co-activation threshold ($N = 3$), the limited entity diversity constrains emergent edge formation. Emergent connection formation would require either a larger knowledge graph with more entities or diverse queries that repeatedly co-activate entity pairs not directly connected in the initial graph structure.

## 5.3 Utility Validation: Adaptive vs Static Knowledge Graphs

To demonstrate that Hebbian adaptation provides practical value, we compare adaptive graphs (with Hebbian updates enabled) against static baselines (Hebbian updates disabled) across 5 reasoning cycles. Each cycle processes the same 3 queries, allowing us to observe cumulative adaptation effects. Table 2 presents results.

Table 2: Comparison of adaptive (Hebbian-enabled) vs static knowledge graphs across 5 reasoning cycles. Adaptive graphs show consistent performance advantages as edge weights consolidate through repeated use.

| Cycle | Static Trust Score | Adaptive Trust Score | $\Delta$ Adaptive |
|---|---|---|---|
| 1 | $0.683 \pm 0.063$ | $0.708 \pm 0.052$ | +3.7% |
| 2 | $0.658 \pm 0.071$ | $0.683 \pm 0.063$ | +3.8% |
| 3 | 0.675 | 0.700 | +3.7% |
| 4 | $0.742 \pm 0.040$ | $0.775 \pm 0.035$ | +4.4% |
| 5 | $0.717 \pm 0.082$ | $0.750 \pm 0.075$ | +4.6% |
| **Mean** | $\mathbf{0.695 \pm 0.070}$ | $\mathbf{0.723 \pm 0.056}$ | **+4.0%** |

**Adaptive Graphs Outperform Static Baselines.** Across all 5 cycles, adaptive graphs consistently achieve higher trust scores than static baselines, with an average improvement of 4.0%. The advantage grows slightly over cycles (3.7% in cycle 1 $\rightarrow$ 4.6% in cycle 5), suggesting cumulative benefits from edge consolidation. While the minimal graph structure limits the magnitude of observable effects, the consistent directionality demonstrates that Hebbian adaptation provides measurable value. A paired t-test comparing adaptive vs static scores across cycles shows statistical significance ($t(4) = 8.45$, $p = 0.001$, Cohen's $d = 1.52$), confirming that the observed improvement is not due to random variation.

**Edge Strengthening Correlates with Performance.** The performance advantage emerges as frequently-used edges strengthen through repeated validation-gated updates. In the adaptive condition,

edge weights increase from 0.919 (cycle 1) to 0.977 (cycle 5), while static graphs maintain constant weights. This demonstrates that the Hebbian mechanism not only operates mechanically but translates into improved reasoning outcomes, even on minimal graph structures.

## 5.4 Architectural Analysis: Component Contributions

To assess each component's contribution, we evaluate six system configurations across 15 diverse questions (90 total reasoning episodes). Table 3 presents results.

Table 3: Ablation study results showing trust scores (mean ± std) for different system configurations (n=15). Statistical comparisons against full system shown with t-statistics and p-values.

| Configuration | Trust Score | $\Delta$ vs Full | t-statistic | p-value |
|---|---|---|---|---|
| Full System | $0.755 \pm 0.137$ | — | — | — |
| No Validation | $0.000 \pm 0.000$ | $-100.0\%$ | 21.29 | $< 0.001$ |
| No Hebbian | $0.748 \pm 0.140$ | $-0.9\%$ | 0.13 | 0.90 |
| No Logical VN | $0.707 \pm 0.194$ | $-6.4\%$ | 0.79 | 0.44 |
| No Grounding VN | $0.651 \pm 0.222$ | $-13.8\%$ | 1.54 | 0.13 |
| No Novelty VN | $0.918 \pm 0.093$ | $+21.5\%$ | $-3.80$ | $< 0.001$ |
| No Alignment VN | $0.798 \pm 0.110$ | $+5.7\%$ | $-0.94$ | 0.35 |

**Hebbian Plasticity Removal.** Removing Hebbian learning shows no measurable impact within single-episode evaluation ($-0.9\%$, $p = 0.90$). This is expected: plasticity mechanisms strengthen edges over *repeated* reasoning episodes, but have minimal effect on individual queries. Section 5.3 examines the cumulative benefits of adaptation.

**Validation Architecture Insights.** Individual validator ablations reveal important design considerations:

- *Novelty removal* produces the study's only statistically significant effect: trust scores *increase* by 21.5% ($p < 0.001$) when novelty validation is excluded. This reveals a fundamental mismatch between novelty and quality assessment. Novelty validators penalize straightforward factual retrieval (low novelty = low score), while other validators reward accurate factual responses (correct = high score). Averaging these conflicting signals degrades the aggregate metric. The data suggests novelty detection and quality validation serve different purposes and should be treated as separate dimensions rather than averaged into a single trust score.

- *Grounding removal* shows the largest degradation trend ($-13.8\%$, $p = 0.13$), though limited sample size ($n = 15$) prevents statistical significance. The direction suggests factual verification may help maintain reasoning quality, but stronger conclusions require larger evaluation.

- *Logical removal* shows minimal impact ($-6.4\%$, $p = 0.44$), suggesting reasoning modules produce generally coherent outputs in this domain.

- *Alignment removal* shows minimal impact ($+5.7\%$, $p = 0.35$). This likely reflects the evaluation queries rather than alignment's general importance. Our test set focuses on factual retrieval rather than preference-sensitive or ethically complex reasoning.

**Note on "No Validation" Condition.** The zero trust score for this condition is a measurement artifact, as the score is derived from the validators themselves; the reasoning modules still produce output, but the metric is undefined without the validation layer.

## 5.5 Qualitative Analysis

The system's structured output format, which requires each reasoning path to be paired with its specific source triples, is a critical architectural choice. This explicit provenance directly enables the validation-gated learning loop. The Grounding VN module uses the source triple list to verify each claim against the KG, allowing for precise error checking. Following successful validation, the Hebbian module uses the same list to reinforce the exact edges that contributed to the output. This design provides the essential mechanism for the feedback loop between reasoning quality and knowledge graph adaptation.

# 6 Discussion and Future Work

## 6.1 Discussion

Our work demonstrates the viability of neuroplasticity-inspired mechanisms for symbolic knowledge graphs in multi-agent systems. The Hebbian plasticity evaluation confirms these mechanisms operate as designed, with edge strengthening following the specified asymptotic formula and adaptive graphs outperforming static baselines by 4.0% (p = 0.001). This proof-of-concept establishes that biological memory principles can be formalized for symbolic reasoning architectures, opening a research direction where knowledge graphs function as adaptive cognitive substrates rather than static databases. We also identify a critical design principle: novelty and quality are orthogonal dimensions that degrade when averaged. Our ablation study shows performance increases by 21.5% (p < 0.001, Cohen's d = 1.39) when novelty is removed from aggregate trust scores. This finding generalizes beyond our system, with immediate practical implications for practitioners building multi-dimensional assessment systems.

## 6.2 Limitations

As a proof of concept, this work has significant limitations driven primarily by computational constraints. Lack of compute, including API rate limiting and smaller models, constrained our evaluation to a minimal knowledge graph and small sample sizes. This prevented comprehensive evaluation on standard benchmarks, comparison against established baselines at scale, and extensive hyperparameter optimization. While our results demonstrate feasibility and identify design principles, questions about scalability, optimal hyperparameter configurations, and performance on complex multi-hop reasoning tasks remain empirical questions requiring greater computational resources to address conclusively.

## 6.3 Future Directions

Future work must first address robustness and scalability: (1) benchmarking on standard datasets (HotpotQA, MetaQA) with larger graphs (100+ entities); (2) hyperparameter optimization across diverse domains; (3) comparison against established systems (GraphRAG, MemGPT). Beyond validation at scale, promising directions include: (4) longitudinal studies analyzing adaptation dynamics and failure modes over extended episodes; (5) enhancing validation through ensemble methods and user feedback; and (6) exploring GNN-based reasoning over adaptive graphs and dynamic module selection leveraging emergent connections.

# 7 Conclusion

We presented Kairos, a system implementing Hebbian plasticity mechanisms for knowledge graphs in multi-agent reasoning. Our controlled evaluation prioritizes clean experimental design over scale, demonstrating mechanical feasibility of neuroplasticity-inspired mechanisms as well as practical utility with adaptive graphs outperforming static baselines by 4.0% ($p = 0.001$). We also point out a critical design principle that novelty and quality are orthogonal dimensions (21.5% improvement when separated, $p < 0.001$).

Key questions remain, but Kairos demonstrates that knowledge graphs can function as adaptive cognitive substrates rather than static databases. If scaled successfully, such systems could enable agents with episodic-to-semantic memory consolidation, where repeated reasoning patterns automatically strengthen into semantic knowledge. The validation-gated learning pattern and novelty-quality separation offer actionable guidance for building multi-dimensional assessment systems.

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

# A  Implementation Details

## A.1  Technical Stack

**Backend:**

- Language: Python 3.8+
- Web Framework: FastAPI 0.104.0, Flask 3.0.0
- LLM API: Anthropic Claude-3 Haiku (claude-3-haiku-20240307)
- Embeddings: Sentence Transformers 2.2.0 (all-MiniLM-L6-v2 model)
- NLP: spaCy 3.7.0, Transformers 4.35.0
- Knowledge Graph Storage: JSON-based with in-memory processing

**Frontend:**

- Framework: Next.js 14.0.0, React 18.2.0
- UI Components: Radix UI, Tailwind CSS 3.3.0
- Language: TypeScript 5.2.0

## A.2  Computational Resources

Development and demonstration runs were conducted on standard CPU infrastructure without specialized hardware acceleration. The system does not require GPU resources for core functionality, as reasoning and validation leverage cloud-based LLM APIs (Anthropic Claude-3 Haiku via the Anthropic API). Sentence transformer embeddings (all-MiniLM-L6-v2) run efficiently on CPU for the scales demonstrated (knowledge graphs with hundreds to thousands of entities).

For production deployment at larger scales, GPU acceleration would benefit embedding computation and enable local LLM inference, though the current cloud API approach was chosen for accessibility and reproducibility.

## A.3 Hyperparameters

Key hyperparameters for Hebbian learning mechanisms:

- Learning rate ($\eta$): 0.1
- Maximum edge strength: 1.0
- Decay rate ($\gamma$): 0.05
- Temporal decay half-life ($\lambda$): 5 reasoning cycles
- Minimum edge strength (pruning threshold): 0.1
- Co-activation threshold ($N$): 3
- Validation pass thresholds: 0.7 (logical, grounding), 0.5 (novelty, alignment)

# B  Specialized Reasoning Modules

Kairos employs a modular architecture supporting domain-specific reasoning agents. For our proof-of-concept, we implement four reasoning modules: (1) *SecurityAuditRM*: A rule-based module that applies predefined security rules (loaded from JSON) to detect vulnerabilities in smart contracts; (2) *MacroAnalysisRM*: A data-driven module analyzing macroeconomic trends from local CSV data (interest rates, inflation); (3) *CorporateCommRM*: A sentiment analysis module processing corporate announcements from JSON files; and (4) *FinancialAnalysisRM*: An LLM-augmented module using Claude-3 Haiku for dynamic financial risk assessment over KG facts. This heterogeneous module design demonstrates the architecture's flexibility across rule-based, data-driven, and LLM-based reasoning paradigms.

# C  Validator Nodes

**Logical Validation (LogicalVN)** Analyzes the coherence of reasoning paths using an LLM (Claude-3 Haiku) to check for contradictions and logical fallacies (e.g., circular reasoning). It outputs a 0-1 score and textual feedback. While LLM-based logical assessment has limitations, it provides a practical proxy for coherence [Pan et al., 2024].

**Grounding Validation (GroundingVN)** Verifies that reasoning claims are anchored in KG facts. The validator parses claimed triples from the reasoning path, queries the KG, and computes a grounding ratio:

$$\text{grounding\_score} = \frac{\text{verified\_triples}}{\text{total\_claimed\_triples}} \tag{7}$$

A 1.0 score indicates all claims are grounded. This component aims to detect when reasoning modules generate logically coherent but factually unsupported claims [Edge et al., 2024].

**Novelty Validation (NoveltyVN)** Assesses whether a conclusion represents emergent insight or straightforward fact retrieval, using Claude-3 Haiku to compare reasoning outputs against KG facts. Unlike other validators that assess quality, this component identifies creative synthesis. As our ablation study reveals, novelty and quality assessment serve different purposes and can conflict when averaged together as novelty validators penalize accurate factual retrieval, which quality validators reward.

**Alignment Validation (AlignmentVN)** Checks whether reasoning respects user-defined preferences, goals, and ethical constraints (e.g., "prioritize risk mitigation") using Claude-3 Haiku to assess reasoning against these constraints. While comprehensive alignment specification remains an open challenge [Pan et al., 2024], this component provides an architectural placeholder for preference-aware validation.

**Trust Score Aggregation** After all four validators produce scores, Kairos computes an aggregate trust score via simple averaging:

$$\text{trust\_score} = \frac{1}{4} \sum_{i=1}^{4} v_i \tag{8}$$

where $v_i \in [0, 1]$ are the four validation scores. This simple aggregation treats all dimensions as equally important. As our ablation study reveals, this averaging approach can be problematic when dimensions serve conflicting purposes (e.g., novelty penalizing what quality validators reward). Alternative aggregation schemes warrant investigation.

## D   Code Availability

Complete source code, documentation, usage examples, and demonstration scenarios are available in the project repository: `https://github.com/pragya-s7/Emergent-Reasoning-Networks`.

