# OpenReview forum: "Validation-Gated Hebbian Learning for Adaptive Agent Memory"
_NeurIPS.cc/2025/Workshop_Mexico_City/NORA — NeurIPS 2025 Workshop NORA Oral_

### Official Review · Reviewer_hvn6 · 2025-11-03
**The official review**

**Rating:** 5
**Confidence:** 3

**Review:**

Summary: The paper introduces a multi-agent reasoning system Kairos, that applies Hebbian neuroplasticity mechanisms to enable adaptive knowledge graph reasoning. The paper presents implementations of three core mechanisms: edge strengthening, temporal decay, and emergent connection formation to improve KG reasoning. The experimental evaluations on the constructed dataset covering five types of reasoning tasks show the superior performance of the proposed adaptive system over the static baseline. The paper also provides a comprehensive ablation study.

Strengths
- The biological inspiration of Hebbian plasticity principles to symbolic knowledge graphs is well-motivated and explained.
- The finding that novelty and quality are orthogonal dimensions that degrade when averaged has immediate practical implications for multi-dimensional assessment systems.
-  The authors thoroughly present the limitations of the proposed approach and provide the mechanical validation to confirm the intended Hebbian mechanisms behavior to support the main contribution.


Weaknesses
- No comparison against established baselines (GraphRAG[1], MemGPT[2], HippoRAG[3]) is provided.
- The ablation study shows that removing Hebbian learning has essentially no impact (Full System vs No Hebbian in Table 3). While the authors explain this is expected for single-episode evaluation, the question remains: if benefits would scale or if the minimal graph structure artificially inflates relative improvements?
-  While four reasoning modules (Security, Macro, Corporate, Financial) are listed, their actual contribution is unclear. The evaluation appears to use a small subset of queries that may not exercise the full multi-agent coordination the architecture supports.
- Figure 1 would benefit from the detailed illustration showing the validation-gated learning feedback loop and Hebbian update process.

[1] Edge, D., Trinh, H., Cheng, N., Bradley, J., Chao, A., Mody, A., ... & Larson, J. (2024). From local to global: A graph rag approach to query-focused summarization. arXiv preprint arXiv:2404.16130.

[2] Packer, C., Fang, V., Patil, S., Lin, K., Wooders, S., & Gonzalez, J. (2023). MemGPT: Towards LLMs as Operating Systems.

[3] Jimenez Gutierrez, B., Shu, Y., Gu, Y., Yasunaga, M., & Su, Y. (2024). Hipporag: Neurobiologically inspired long-term memory for large language models. Advances in Neural Information Processing Systems, 37, 59532-59569.

---

### Official Review · Reviewer_sNXN · 2025-11-04
**Review of paper “Validation-Gated Hebbian Learning for Adaptive Agent Memory”**

**Rating:** 6
**Confidence:** 3

**Review:**

This paper proposes a new computation model of Hebbian-based learning for Knowledge Graphs (which leads to dynamic KGs) embedded in a multi-agent system comprising of dedicated reasoning modules and a four-dimensional validator which assesses the logical consistency, grounding, novelty, and alignment of each candidate update to a target KG.

Pros:

1)	This is a very interesting paper showcasing a biological-inspired method for updating the strengths in the edges of KGs, and forming new connections based on the frequency of entities co-occurring in input contexts.
2)	The specific mechanisms proposed for this model seem faithful to theory in the neuroscience of learning, more specifically, they seem aligned to Hebbian learning methods.
3)	Relevant dimensions for evaluating the proposed model are proposed such as assessing the validity of the mechanisms, a comparison with a static knowledge graph, and some ablation studies.
4)	Code is released.

Weaknesses:

1)	What I believe to be the main weakness is the highly limited evaluation of the proposed model by using an overly simplified dataset and knowledge graph.
2)	There is an unclear (and critical) element in the evaluation setup: in Sections 5.1 and 5.2 it is mentioned that a set of 3 queries is used for evaluation; however, it is not clear if only 3 queries are used for the whole evaluation (leading to the results shown in Tables 1 and 2), which would highly tamper with obtaining a sound conclusion of the validity of the proposed mechanisms, or sets of 3 queries are drawn from the dataset and are averaged, which would be a restricted evaluation but a more reasonable scenario.

Minor weaknesses and suggestions:

1)	Figure 1 is not cited in the main text.
2)	It would be nice to see an interpretation of the meaning of the Trust Scores shown in Tables 1 and 2
3)	It would also be nice to have a qualitative example of how the Hebbian-based learning takes place in the knowledge graph
4)	Ideally, the paper should elaborate on the dedicated reasoning modules: what is their role in the multi-agent system? Otherwise, we are left in a sort of vacuum

---

### Official Review · Reviewer_YcxN · 2025-11-05
**Highly Novel Architecture for Adaptive Agent Memory, Requires Scale-Up and Design Principle Refinement**

**Rating:** 6
**Confidence:** 4

**Review:**

1. Novelty
This paper presents a highly novel and inspiring architecture in the field of LLM agent memory, primarily by introducing Hebbian plasticity mechanisms into symbolic Knowledge Graphs (KGs) to enable self-adaptation and structural reinforcement based on reasoning success. The most significant innovation is the Validation-Gated Learning mechanism, which is a major advancement over existing static KG memory approaches, as it directly addresses the critical problem of preventing erroneous reasoning paths, such as hallucinations, from being reinforced and solidified into the agent's memory. While the Hebbian principle itself is not entirely new, the authors should more explicitly clarify in the introduction and related work sections how their approach differs from existing GNN-based Hebbian methods or classical cognitive architectures like ACT-R, emphasizing the uniqueness of the "validation gate" and "symbolic structural adaptation".

2. Clarity
The paper is well-structured and logically sound, clearly moving from the problem context (memory challenges in LLM agents) to the proposed solution (the Kairos system). The abstract and introduction effectively summarize the system's innovations and contributions, and Section 3 (System Architecture), along with Figure 1, provides a clear explanation of the complex feedback loop. The paper's key finding—the orthogonality of novelty and quality—is well-supported by the ablation study and clearly discussed. However, while the mathematical formulas for the Hebbian mechanisms (Eq. 1-5) are clear, the surrounding text should offer more intuitive explanations for the chosen values of hyperparameters like η (learning rate) and γ,λ (decay parameters), detailing, for example, why η=0.1 is chosen to prevent "single-trial over-consolidation".

3. Reproducibility
The paper provides sufficient information for the system's core design, including the detailed system architecture, the technical stack (leveraging Anthropic Claude-3 Haiku and Sentence Transformers), and the exact numerical values for key hyperparameters. Furthermore, the authors state that the source code is available. However, the authors explicitly acknowledge that the evaluation was constrained by limited computational resources, forcing them to use a minimal knowledge graph and small sample sizes, which prevented comprehensive evaluation on standard LLM benchmarks (like HotpotQA or MetaQA). To maximize the reproducibility of the limited-scale experiments, the authors are strongly encouraged to provide a more detailed appendix that describes the structure of the minimal KG and the construction of the evaluation dataset.

4. Ethical Compliance
The paper demonstrates strong ethical compliance. One of the core system components, the Alignment Validation (AlignmentVN) agent, is specifically designed to check if the reasoning respects user-defined preferences and ethical constraints. This directly addresses potential misuse and ethical considerations. Moreover, the Validation Gate itself functions as a crucial safety mechanism by preventing the system from learning and solidifying inaccurate or potentially harmful knowledge, thereby mitigating the risk of hallucination reinforcement.

---

### Official Review · Reviewer_aRwY · 2025-11-07
**Review for Validation-Gated Hebbian Learning for Adaptive Agent Memory**

**Rating:** 7
**Confidence:** 4

**Review:**

## Paper Summary

This paper presents Kairos, a multi-agent reasoning system that implements Hebbian plasticity mechanisms for adaptive knowledge graphs. It leverages concepts from neuropsychological theory about plasticity of neurons and formalized three operations: edge strengthening, temporal decay, and emergent connection formation. Kairos is validated through three complementary experiments, mechanical validation, utility validation, and architectural analysis. Experiments provides a proof-of-concept support for Kairos.


## Strengths

1. This paper provides a conceptually connection between neuroplasticity and symbolic AI, providing a new direction for multi-agent validation loop. It demonstrates how to leverage neuroscience-inspired learning in modern LLM and agent applications.
2. Despite being a proof-of-concept, the experiments demonstrate both mechanical correctness (edge strengthening, decay) and functional improvement over static KGs.

## Weaknesses

1. Experiment dataset is too small, limiting its support for wider validation of the proposed method.
2. The proposed method is heuristic without strict justification, lacking a stronger theoretical support.

---

### Official Review · Reviewer_YtsR · 2025-11-07
**My review of "Validation-Gated Hebbian Learning for Adaptive Agent Memory"**

**Rating:** 7
**Confidence:** 4

**Review:**

The authors of this paper describe Kairos, a novel multi-agent reasoning architecture inspired by biological memory.  This approach to adaptive memory is modeled by multiple specialized sub-agent modules (leveraging an Anthropic LLM) operating on a dynamic knowledge graph with three Hebbian-inspired plasticity operations: edge strengthening, temporal decay, and emergent connection formation. These operations are gated by a multi-dimensional validation layer (logical, grounding, novelty, alignment).  The authors claim the following four contributions:

1. This architecture prevents hallucination reinforcement while enabling adaptive memory
2. Formalization of three neuroplasticity-inspired mechanisms for symbolic KG structures
3. A multi-agent validation framework for four quality dimensions (logical consistency, factual grounding, novelty, alignment).
4. POC validating design choices yield both mechanical correctness and practical utility, with adaptive graphs outperforming static baselines

This reviewer really enjoyed reading this paper.  My highlights for this paper were the following:
- Novel architectural idea: Really innovative concept! The validation-gated feedback loop between reasoning quality and KG adaptation was well-motivated.  Contribution claim 3, check.
- Clear and well-structured writing and presentation:  At first glance the title was intimidating but it was actually a joy to read.
- Concrete formalization of plasticity: The authors provided formulas for all three operations.  Contribution claim 2, check.
- Useful design insight: The POC evaluation demonstrates that novelty and quality are orthogonal
- Great Reproducibility: The authors provide a GitHub repo with excellent MD instructions, tech stack description, and hyperparameter values.

Here is my list of opportunities to improve this work:

- This reviewer could not find direct evidence in this work that explicitly substantiates claim 1.
- As noted in the limitations section, the evaluation scale is too small. I understand it is an early POC and agree with the authors suggestions for future work including expanding evaluation to established benchmark so they can strengthen claim 4.
- Also noted, the emergent connections operation was not validated.  This was disappointing because it's a concept with interesting potential.
- How many calls to the LLM were required per query? How many tokens? Cost?  It would be nice if the authors provided this information.
- Would appreciate the authors report the per-dimension score for validators. Perhaps these should be primary?  Likewise for including Validation Pass-Fail rates.
- Although the Architectural diagram was helpful, it could be enhanced with a second sequence diagram walking through a tangible example of one of the queries.
- Small nitpick: I really appreciate it when prompts are provided in the appendix.  If it didn't fit, perhaps extracting the prompts out of the code in your repo into separate text files would be a helpful way for reviewers and future contributors to ramp up quickly.

This reviewer did not find any ethical issues with this work.

Overall, this reviewer really enjoyed the paper.  After reading the eval section, I was left hungry for more. More evidence.  More experiments.  Stronger comparisons to existing methods to back up claims.   Typically given the early stage of this work, I would reject for Main Conference.  But because this is for a workshop in an emerging space and our instructions included: "we aim to select papers with novelty and solid contributions while maintaining a degree of tolerance for early-stage or exploratory work" I have decided to accept.  The concept is novel, relevant, well documented, and appears to be reproducible.  I look forward to future updates from the authors on this work.

---

### Official Review · Reviewer_JcHP · 2025-11-07
**##1. Summary: The paper proposes **Kairos**, a multi-agent reasoning system integrating **Hebbian plasticity** into **knowledge graphs (KGs)**. The triggering of structural adaptation by reasoning outcomes is only possible when these outcomes are validated across four dimensions: logical coherence, factual grounding, novelty, and alignment. This validation process serves to prevent the reinforcement of hallucinations. Main claims: The following four points constitute the foundation of the present study: firstly, the formalisation of neuroplasticity-inspired mechanisms; secondly, the validation-gated adaptation loop; thirdly, the proof-of-concept showing that adaptive graphs outperform static ones by approximately 4%; and fourthly, the identification that novelty and correctness are **orthogonal** metrics.**

**Rating:** 8
**Confidence:** 3

**Review:**

## 2. Evaluation

### **A. Quality**

Mechanisms are **mathematically sound** and internally consistent.

[
\Delta s = \eta (s_{\max} - s_{\text{current}}), \quad
s' = \min(s_{\max}, s_{\text{current}} + \Delta s)
]

[
\text{decay} = \gamma (1 - e^{-c/\lambda}), \quad
s' = \max(s_{\min}, s_{\text{current}} - \text{decay})
]
Empirical results (e.g., (t(4)=8.45,, p=0.001)) confirm proper functioning. Yet, evaluation uses small synthetic data, limiting external validity.
**Conclusion:** solid methodology, modest scope.

### **B. Clarity**
The paper is well-structured and readable, with clear architecture and notation. I would recommend that overly assertive phrasing be moderated.
Conclusion: The document is characterised by its clarity and technical transparency.

### **C. The originality
The introduction of validation-gated Hebbian adaptation for symbolic reasoning represents a novel approach. The distinction between novelty and quality metrics is conceptually sound and of value in this context.
Conclusion: The work displays a high degree of conceptual originality.

### **D. Significance:
The work's impact is principally **directional** in nature, indicating feasibility rather than scale. It is recommended that future benchmarking on datasets such as HotpotQA or MetaQA be undertaken in order to strengthen claims.
Conclusion: This foundation demonstrates considerable potential for further research in the field of adaptive agent memory.

## 3. Pros and Cons

| **Pros**                                             | **Cons**                                                             |
| ---------------------------------------------------- | -------------------------------------------------------------------- |
| Novel biologically inspired adaptation mechanism     | Evaluation limited to toy graphs                                     |
| Validation gate prevents hallucination reinforcement | No large-scale or baseline comparison                                |
| Transparent modular design                           | Simplistic averaging of validation scores (T=\tfrac{1}{4}\sum_i v_i) |
| Rigorously formulated learning rules                 | Conceptual overreach in cognitive analogies                          |
| Clear insight: novelty ≠ correctness                 | Novelty metric penalizes factual retrieval                           |


## 4. Suggestions

1. Benchmark on standard QA datasets to test scalability.
2. Refine trust-score aggregation beyond simple averaging.
3. Integrate GNN-based reasoning for richer adaptation.
4. Temper analogical claims to biology; emphasize computational aspects.

---

### 5 **Final Remark**

*Validation-Gated Hebbian Learning* offers an elegant bridge between **biological memory models** and **symbolic AI**. Despite limited scale, it sets a compelling precedent for future research on **adaptive, self-organizing cognitive architectures**.